# Activity of Total Alcohol Dehydrogenase, Alcohol Dehydrogenase Isoenzymes and Aldehyde Dehydrogenase in the Serum of Patients with Alcoholic Fatty Liver Disease

**DOI:** 10.3390/medicina58010025

**Published:** 2021-12-24

**Authors:** Blanka Wolszczak-Biedrzycka, Elżbieta Zasimowicz-Majewska, Anna Bieńkowska, Grzegorz Biedrzycki, Justyna Dorf, Wojciech Jelski

**Affiliations:** 1Department of Psychology and Sociology of Health and Public Health, University of Warmia and Mazury in Olsztyn, Warszawska 30, 10-082 Olsztyn, Poland; anna.bienkowska@uwm.edu.pl; 2Medical Diagnostic Laboratory, Pro-Medica Hospital in Elk, Baranki 24, 19-300 Elk, Poland; majewskaela8704@gmail.com; 3Hospital Dispensary, Regional Specialist Hospital in Olsztyn, Żołnierska 18, 10-561 Olsztyn, Poland; gbiedrzycki@wss.olsztyn.pl; 4Department of Laboratory Diagnostics, Medical University in Bialystok, Waszyngtona 15a, 15-269 Bialystok, Poland; justyna.dorf@umb.edu.pl; 5Department of Biochemical Diagnostics, Medical University in Bialystok, Waszyngtona 15a, 15-269 Bialystok, Poland; wjelski@umb.edu.pl

**Keywords:** alcohol dehydrogenase isoenzymes, aldehyde dehydrogenase, ADH, ALDH, alcoholic fatty liver

## Abstract

*Background and objectives*: The aim of the current study was to assess the use of determinations of total alcohol dehydrogenase and the activity of its isoenzymes as well as aldehyde dehydrogenase in the serum of patients with alcohol liver disease. *Materials and Methods*: The testing was performed on the serum of 38 patients with alcoholic fatty liver (26 males and 12 females aged 31–75). The total activity of ADH was determined by the colorimetric method. The activity of ADH I and ADH II, as well as ALDH, was determined by the spectrofluorometric method using fluorogenic specific substrates. The activity of isoenzymes of other classes was determined by spectrophotometric methods using substrates. *Results*: A statistically significantly higher ADH I activity was noted in the serum of patients with alcoholic fatty liver (4.45 mIU/L) compared to the control group (2.04 mIU/L). A statistically significant increase in the activity was also noted for the class II alcohol dehydrogenase isoenzyme (29.21 mIU/L, control group: 15.56 mIU/L) and the total ADH (1.41 IU/L, control group: 0.63 IU/L). *Conclusions*: The obtained results imply the diagnostic usefulness of the determination of AHD total, ADH I, and ADH II activity in the serum of patients with alcoholic fatty liver.

## 1. Introduction

Fatty liver is defined as a process of increased accumulation of lipid compounds in the hepatocytes. Normally, fat accounts for 3–5% of the liver weight. Exceeding this range by more than 5% leads to a condition referred to as fatty liver (FL) [1], which is among the most common liver damage syndromes of an alcohol etiology. The liver is the main organ that metabolizes ethyl alcohol through the oxidative pathway involving alcohol dehydrogenase (ADH) and aldehyde dehydrogenase (ALDH). Even a few days of alcohol abuse may result in steatotic lesions in the liver [2]. Steatosis, being the earliest stage of alcohol-related liver disease, develops very rapidly and can emerge as early as after 3–7 days of excessive alcohol consumption. A 400 g portion of pure alcohol results in fat vacuoles of lesser or greater size accumulating in the hepatocytes. On the other hand, in alcoholics who consume an average of approx. 80 g of pure ethanol on a daily basis for five years, the risk of steatosis reaches almost 80%. The threshold hepatotoxic dose for males amounts to 60–80 g of pure alcohol per 24 h, while for females it amounts to 20–40 g [3]. A complete diagnosis of liver steatosis should include an interview, diagnosis of clinical symptoms, the performance of laboratory testing, ultrasound procedure, and CT or MRI scans. While a liver biopsy is regarded as a gold standard in the diagnosis of liver steatosis, less invasive markers of the disease are still being sought [4].

The presence of mainly total ADH and class I, II, and III ADH isoenzymes as well as ALDH was demonstrated in hepatocytes. Many studies have demonstrated significant changes in the activity of alcohol dehydrogenase (ADH), its isoenzymes, and aldehyde dehydrogenase (ALDH) in the blood serum of patients with different liver diseases, which provides grounds for considering these enzymes as hepatocellular damage indicators [3]. An increase in the activity of ADH and its isoenzymes was demonstrated in the course of, for example, C and B viral hepatitis, liver cirrhosis, liver cancer, and autoimmune hepatitis [5,6,7]. Some authors observed an increase in the activity of total ADH and its isoenzymes (mainly ADH I and ADH II) in the serum of patients suffering from a non-alcoholic fatty liver [8]. Moreover, it is believed that the level of enzyme serum activity is determined by the degree of hepatocellular damage due to the toxic effects of ethyl alcohol [6]. The current study assessed the changes in the activity of alcohol dehydrogenase, the activity of its isoenzymes, and aldehyde dehydrogenase and their diagnostic utility in the serum of patients with alcoholic liver disease.

## 2. Materials and Methods

The research protocol was approved by the Human Care Committee of the Medical University in Bialystok, Poland (Approval No. R-I-002/181/15 March 2014). After a detailed explanation of the purpose of our research and the possible risk, all the qualified patients agreed in writing to participate in the experiment.

### 2.1. Materials

The study material was the blood serum collected for routine biochemical testing from 38 patients aged 31–75 (26 males aged 31–73, and 12 females aged 35–75) with diagnosed alcoholic fatty liver, hospitalized at the Department of Infectious Diseases and Hepatology, Medical University of Białystok, Poland. Diagnosis was performed on the basis of clinical and laboratory investigations (e.g., transaminases activities). Exclusion criteria in the study group included chronic liver disease and hepatic viral infection (HAV, HBV, or HCV) or high BMI > 25.0. All patients had BMI within the normal value (18.5–24.9). All patients had abused alcohol more than 10 years (26 males: over 20 g per day, 12 females: over 10 g per day). None of them had stopped drinking before the hospitalization. Histopathological testing on liver biopsies assessed the steatotic lesions and the degree of cirrhosis and the patients were then classified according to the METAVIR classification, which is used to assess the severity of liver fibrosis. The fibrosis stage can be graded on a five-point scale: F0—no fibrosis, F1—expansion of portal zones, F2—expansion of most portal zones and occasional bridging, F3—expansion of most portal zones and marked bridging, F4—cirrhosis (1st degree was diagnosed in 12 patients, 2nd degree was exhibited by eight patients, 13 patients had 3rd degree, and 4th degree was observed in five patients) [9].

The control group included 40 healthy subjects aged 30–70 (20 males aged 33–70, and 20 females aged 30–65), from whom blood was collected during a routine examination. None in the control group exhibited abnormalities indicative of digestive system (liver, pancreas, the gastrointestinal tract) diseases in the physical examination and laboratory testing results. All subjects in the control group only consumed alcohol occasionally. The number of patients in the study and the control group was set based on a previously conducted pilot study. The power of the study was set at 0.9.

### 2.2. Blood Collection

Fasting venous blood (10 mL) was collected from all patients on empty stomach and upon overnight rest. A vacutainer blood collection system was used for this. Blood was centrifuged at 1500× *g* for 10 min at +4 °C (MPW 351, MPW Med. Instruments, Warsaw, Poland) and the top layer (serum) was taken. Until determinations, all samples were stored at −80 °C.

### 2.3. Determination of Total ADH Activity

Alcohol dehydrogenase is a catalyst of the reduction reaction of p-nitroso-N-N-dimethylaniline (NDMA) involving NADH that is formed during the enzymatic oxidation of n-butanol in the presence of NAD. The amount of enzymatic NDMA reduction, measured at a wavelength of 440 nm, is the measure of enzyme activity.

The reaction mixture (2 mL) comprised 1.9 mL of NDMA (concentration of 25 μmol/L) dissolved in a 0.1 mol/L of sodium phosphate buffer (pH 8.5), and 0.1 mL of a mixture comprising n-butanol (concentration of 0.25 mol/L) and NAD (concentration of 5 mmol/L). The reaction was initiated by the addition of 0.1 mL of serum. The reference mixture (2 mL) comprised the same components plus 4-methylpyrazole (concentration of 12 mmol/L), i.e., a specific ADH inhibitor. After a 20 min incubation (at a temperature of 25 °C), 50 μL of 4-methylpyrazole (concentration of 0.5 mol/L) was added to the study sample. The absorbance of both reaction and reference mixture was measured using an Epoll 20 spectrophotometer at a wavelength of 440 nm. The difference in absorbance between the reference sample and the study sample was then calculated. The difference was converted into the activity expressed in IU/L using the calibration curve provided by the method authors [10].

### 2.4. Determination of ADH I and ADH II Isoenzyme Activity

Alcohol dehydrogenase, with the involvement of NADH, is a catalyst of the reduction reaction of polyaromatic aldehyde specific for a particular isoenzyme class: 4-methoxy-1-naphthaldehyde (ADH I) and 6-methoxy-2-naphthaldehyde (ADH II). The reaction products are respective polyaromatic alcohols that exhibit strong fluorescence, whose intensity is the measure of enzyme activity.

The reaction mixture (3 mL) comprised 150 μL of 4-methoxy-1-naphthaldehyde (used to determine the ADH I activity) or 6-methoxy-2-naphthaldehyde (used to determine the ADH II activity) with a concentration of 300 μmol/L, 100 μL NADH (concentration of 1 mmol/L), and 2.69 mL of sodium phosphate buffer with a pH of 7.6 (concentration of 0.1 mol/L). The reaction was initiated by the addition of 60 μL of blood serum. Changes in fluorescence were recorded after ten minutes with a Shimadzu RF-5301 (PC) spectrofluorometer with an excitation wavelength of 316 nm and an emission wavelength of 370 nm (class I), 360 nm (class II). After the addition of 60 μL of 4-methoxy-1-naphthalenemethanol (for ADH I) or 6-methoxy-2-naphthalenemethanol (for ADH II), i.e., the internal standard, the fluorescence was measured [11].

### 2.5. Determination of ADH III Isoenzyme Activity

Class III isoenzyme of alcohol dehydrogenase is a catalyst of the oxidation reaction of caprylic acid with the involvement of nicotinamide adenine dinucleotide, which is reduced to NADH. A measure of the isoenzyme activity is the NADH formation rate.

The reaction mixture (2 mL) comprised 31 μL of caprylic alcohol (concentration of 1 mmol/L), 240 μL of NAD (concentration of 1.2 mmol/L), and 1629 μL of glycine buffer with a pH of 9.6 (concentration of 0.1 mol/L). The reaction was initiated by the addition of 100 μL of serum. After 10 min, changes in absorbance were recorded using an Epoll 20 spectrophotometer (wavelength of 340 nm, temperature of 25 °C) [12].

### 2.6. Determination of ADH IV Isoenzyme Activity

Class IV ADH isoenzyme is a catalyst of the reduction reaction of m-nitro-benzaldehyde, with the involvement of a reduced form of nicotinamide adenine dinucleotide being oxidized to NAD. A measure of the isoenzyme activity is the NADH decay rate.

The reaction mixture (2 mL) comprised 132 μL of m-nitro-benzaldehyde (concentration of 80 μmol/L), 172 μL of NADH (concentration of 86 μmol/L), and 1596 μL of sodium phosphate buffer with the pH of 7.5 (concentration of 0.1 mol/L). The reaction was initiated by the addition of 100 μL of serum. After 10 min, changes in absorbance were recorded using an Epoll 20 spectrophotometer (temperature of 25 °C, wavelength of 340 nm) [13].

### 2.7. Determination of Total ALDH Activity

Aldehyde dehydrogenase is a catalyst of the oxidation reaction of polyaromatic aldehyde (6-methoxy-2-naphthaldehyde) with the involvement of NAD. The resulting product is 6-methoxy-2-naphthoic acid exhibiting strong fluorescence whose intensity is the measure of enzyme activity.

The reaction mixture (3 mL) comprised 60 μL of 6-methoxy-2-naphthaldehyde with a concentration of 300 μmol/L, 20 μL of NAD with a concentration of 1 mmol/L, and 2.80 mL of sodium phosphate buffer (pH 8.5) with a concentration of 0.1 mol/L. The reaction was initiated by the addition of 60 μL of serum. Changes in fluorescence were recorded after eight minutes with a Shimadzu RF-5301 (PC) spectrofluorometer (excitation wavelength of 310 nm, emission wavelength of 360 nm). A re-measurement of fluorescence was carried out following the addition of 60 μL of 6-methoxy-2-naphthoic acid as the internal reaction standard [14].

### 2.8. Statistical Calculations

A statistical analysis of the obtained results was conducted using Statistica PL software, ver. 10. As the normal distribution was not confirmed by the χ2 test, an analysis of the statistical variability of the differences between the groups being compared was achieved using a non-parametric Wilcoxon test. The values of *p* < 0.05 were assumed to be a statistically significant difference.

## 3. Results

### 3.1. Assessment of ADH and Its Isoenzymes and ALDH Activity in the Serum of Patients with Alcoholic Fatty Liver and in the Control Group

An increase in the total ADH activity was demonstrated in the group of patients with alcoholic fatty liver (1.41 IU/L, respectively) in relation to the activity in healthy subjects (0.63 IU/L) (Table 1, Figure 1). The differences between both patient groups and the control group were statistically significant (*p* < 0.05). Analysis of the total ALDH activity in the serum of patients with alcoholic fatty liver revealed no statistically significant differences compared to the control group.

Analysis of the activity of ADH isoenzymes in the serum of patients with alcoholic fatty liver demonstrated that the highest activity is obtained by the class II ADH. In patients with alcoholic fatty liver, its activity amounted to 29.21 mIU/L and exhibited a statistically significant difference compared to that of the control group (15.56 mIU/L). A significant difference in the serum activity between the patient groups and the healthy subject group was also noted for the class I isoenzyme of alcohol dehydrogenase (patients with alcoholic fatty liver vs. control group, 4.45 mIU/L vs. 2.14 mIU/L, respectively). The ADH III and ADH IV isoenzymes exhibited higher activity in the serum of the study groups patients but there were no statistically significant differences compared to the control group.

### 3.2. Comparison of the Activity of ADH and Its Isoenzymes and ALDH Depending on the Degree of Hepatic Tissue Cirrhosis

Total alcohol dehydrogenase demonstrated a statistically significant increase in the serum activity of patients in each of the four degrees of liver cirrhosis lesion progression compared to the control group. Analysis of the activity of aldehyde dehydrogenase in the serum of patients with alcoholic fatty liver showed no statistically significant differences compared to the control group, irrespective of the degree of liver tissue cirrhosis.

Analysis of the activity of alcohol dehydrogenase isoenzymes in the serum of patients with fatty liver, depending on the degree of hepatic tissue cirrhosis, showed statistically significant higher activity of class I and II ADH isoenzymes in each study group (degrees 1–4) compared to the control group (Table 2, Figure 2). The activity of the studied class III and IV isoenzymes of alcohol dehydrogenase increased with the degree of hepatocellular damage in relation to the control group. However, the differences were not statistically significant.

### 3.3. ROC Analysis

ROC analysis was performed in the study to assess the diagnostic value of alcohol dehydrogenase, ADH I, and ADH II in the diagnostics of alcoholic fatty liver disease (Table 3, Figure 3). We demonstrated that all determined parameters differentiate patients with alcoholic fatty liver disease from healthy controls to a moderate extent. AUC for ADH I was 0.784 with cut-off value >5.05 mIU/L, 74.56% sensitivity and 78.35% specificity; for ADH II was 0.758 with cut-off value >36.81 mIU/L, 70.15% sensitivity and 75.72% specificity; for ADH total was 0.706 with cut-off value >2.45 U/L, 66.42% sensitivity and 70.33% specificity (Table 3, Figure 3).

## 4. Discussion

Almost 40% of liver diseases are caused by alcohol consumption, and fatty liver (or liver steatosis) is the most common of them. It is extremely important to diagnose fatty liver and introduce a therapy quickly enough to considerably reduce pathological lesions in the liver, and to prevent its irreversible damages. What is important in the entire process of diagnosis, treatment, and control of fatty liver is the development of diagnostics especially searching for new, specific, and non-invasive fatty liver markers. It should be aimed at assisting in generating a correct diagnosis, identifying its etiology, and monitoring the course of the disease, as well as enabling the prediction of further consequences of this process.

In the presented study, we assessed as a marker of hepatocellular damage in the course of liver steatosis, the activity of alcohol dehydrogenase, its isoenzymes, and aldehyde dehydrogenase in the serum of patients with alcoholic fatty liver. ADH comprises a family of enzymes that have been grouped into IV classes. Class I is a classical liver ADH but is also detected in the gastrointestinal tract. Class II ADH in humans is found only in the liver, whereas class III is present in all tissues examined to date. Class IV isoenzyme of ADH is preferably expressed in the upper part of the digestive tract. In humans, liver ALDH exists as several isoenzymes which differ in subcellular location [6]. The study showed the greatest increase in the class II ADH isoenzyme, a slightly smaller increase in the class I ADH isoenzyme, and an increase in total ADH in the serum of patients with alcoholic fatty liver in relation to the control group. Jelski et al. evaluated the activity of ADH and its isoenzymes as well as ALDH in the serum of patients with B and C viral hepatitis and non-alcoholic fatty liver [8,15,16]. They observed a significant increase in the activity of class I and II ADH isoenzyme in viral hepatitis C and non-alcoholic fatty liver.

However, for hepatitis B, the results showed an increase in the activity of class I and II ADH in the serum of the study group, in comparison with the control group. An increase in the activity of class II ADH isoenzyme in patients with fatty liver and hepatitis C and B may associate with hepatic cell damage and depend on the degree of its damages. Since ADH II is mainly located in the hepatocyte mitochondrial matrix, the greater the damage to the cell, the greater the increase in its activity. High ADH I activity in the serum of the study group may also indicate the hepatic origin of the isoenzyme. It was demonstrated that the activity of class I isoenzyme of alcohol dehydrogenase, measured by the spectrofluorometric method, can be a useful marker of liver damage in the course of severe hepatitis B and chronic hepatitis (irrespective of the alcoholic or viral etiology) [16,17].

The presence of alcohol dehydrogenase is found in both healthy organ cells and pathologically altered (by neoplasms or inflammation) tissues. Changes in the activity of individual ADH isoenzymes may cause many metabolic disorders and consequently result in the intensification of the disease process. Importantly for laboratory diagnostics, alcohol dehydrogenase can be released to the blood from pathologically altered cells. As is the case with many other indicator enzymes, a considerable increase in the activity of total ADH or its individual isoenzymes in the serum of patients can be observed. Numerous studies have been conducted into the ADH activity in the blood serum of patients with gastrointestinal tumors: colorectal, gastric, liver, and pancreatic cancers [6,18,19,20]. It has been demonstrated a statistically significantly increased activity of this enzyme as compared to the blood serum of healthy subjects. It is well known that the liver is particularly exposed to the toxic and carcinogenic effects of ethanol because it is the main source of ADH—a major enzyme system responsible for the oxidation of ethanol. It was demonstrated that in neoplastic hepatic cells, the activity of class I ADH isoenzyme is almost 25% higher than that in normal hepatic cells [18]. An increase in the activity of ADH I demonstrates a positive correlation with the total activity of alcohol dehydrogenase. A very high level of ADH I, particularly in the serum of patients with metastatic liver cancer, may be due to either the release of this isoenzyme from the hepatic neoplastic cells or from the primary neoplastic focus located in another organ [6]. It was demonstrated that in people abusing alcohol, a change in the activity of this isoenzyme was proportional to the degree of organ damage. However, Chrostek et al. [17] pointed out that it could be of non-hepatic origin and originate from the ethanol-damaged cells of gastric mucosa and the intestines. Recent reports have also noted the class II ADH isoenzyme as a potential prognostic parameter for hepatocellular carcinoma [21]. It seems extremely important that changes in ADH activity may be a useful parameter in monitoring the effectiveness of treatment of alcohol-related liver disease. So far, it has been demonstrated that there is a link between ALDH activity and liver damage, especially with alcohol etiology. According to Enomoto et al. [22], the population most exposed to liver damage due to excessive alcohol consumption includes persons in whom acetaldehyde is quickly formed with the involvement of the efficient alcohol dehydrogenase system with the slowed elimination of this compound by inefficient ALDH2. Almost 50% of Asian people have a genetically determined ALDH defect which leads to the formation of toxic levels of acetaldehyde following the consumption of even a small portion of alcohol. This promotes the development of alcohol-related liver disease and, consequently, fatty liver [23]. It could be supposed that the high activity of ADH influences the possible development of the disease, as it happens with liver cancer [6].

It is a noteworthy fact that the information derived from medical history taking is not always able to help clearly indicate the background of liver steatosis, particularly for an alcohol etiology (patients often tend to conceal their actual excessive alcohol consumption). In this case, biochemical tests are helpful. Particularly important is the ratio of aspartate transaminase to alanine transaminase, which, in the early period of the alcoholic form, is greater than 2:1, while in the non-alcoholic form, ALT is usually predominant [4,22]. Elevated GGT activity can be observed more frequently in the alcoholic fatty liver than in the non-alcoholic liver [24]. Despite the development of biochemical tests, it should be noted that an increase in the activity of individual parameters is not specific for liver steatosis. the tests may be helpful in the assessment of the degree of liver damage. Great emphasis should be placed on searching for new markers of fatty liver, with high specificity and relatively low costs of determination. It is worth noting that both alcohol dehydrogenase and aldehyde dehydrogenase, which are the subject of the presented study, are not homogeneous enzymes but isoenzymes complexes characterized by, e.g., different tissue localization [4].

Of all the study enzymes, class I and II ADH and total ADH isoenzymes showed the greatest diagnostic usefulness. Statistically significant higher activity of these parameters was noted in the serum of patients with alcoholic fatty liver disease, in relation to the control group. Similar results were obtained in a study performed by Jelski et al. assessing ADH isoenzyme activity depending on the degree of liver cirrhosis in patients with non-alcoholic fatty liver disease. It has been demonstrated higher activity of class I and II ADH isoenzymes in each study group (degrees 1–4 of cirrhosis) in comparison with the control group. Total alcohol dehydrogenase also showed a statistically significant increase in activity in the serum of patients in each of the four degrees of liver cirrhosis lesion progression compared to the control group [7]. Moreover, we observed the increased activity of total ADH activity and its isoenzymes (ADH I, ADH II), which correlate with the level of fatty liver, but the difference was not statistically significant.

## 5. Conclusions

This study identified the possibilities for the application of alcohol dehydrogenase (and its isoenzymes) as well as aldehyde dehydrogenase in the diagnosis of liver steatosis. Based on the study results, it was demonstrated that the considerable increase in ADH II, ADH I, and ADH total activity in the blood serum of patients with fatty liver is caused by the release of these isoenzymes from the liver cells altered by steatosis. Assessment of the diagnostic usefulness indices implies the possibility of applying the class I and II isoenzyme as well as total ADH as a marker of liver steatosis.

## Figures and Tables

**Figure 1 medicina-58-00025-f001:**
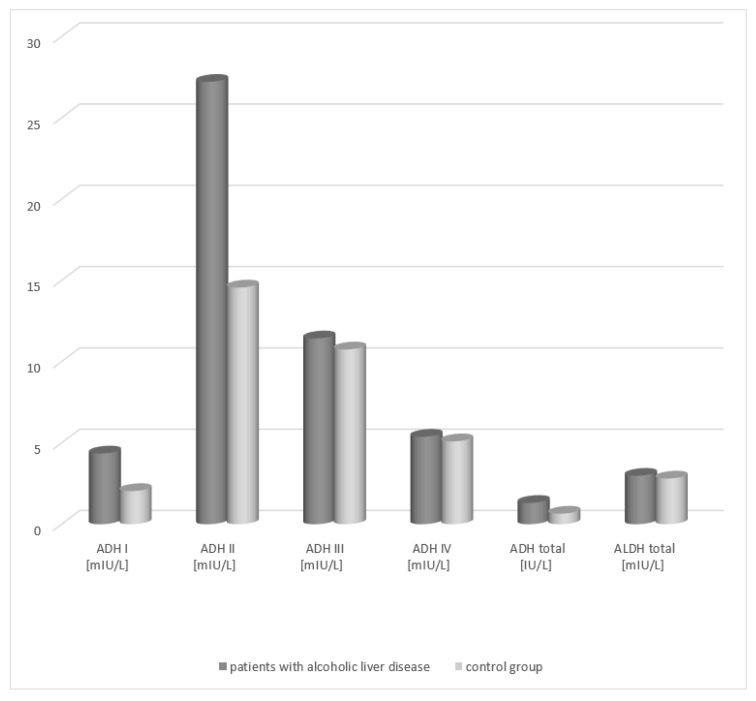
Activity of ADH and ALDH in the serum of patients with alcoholic fatty liver.

**Figure 2 medicina-58-00025-f002:**
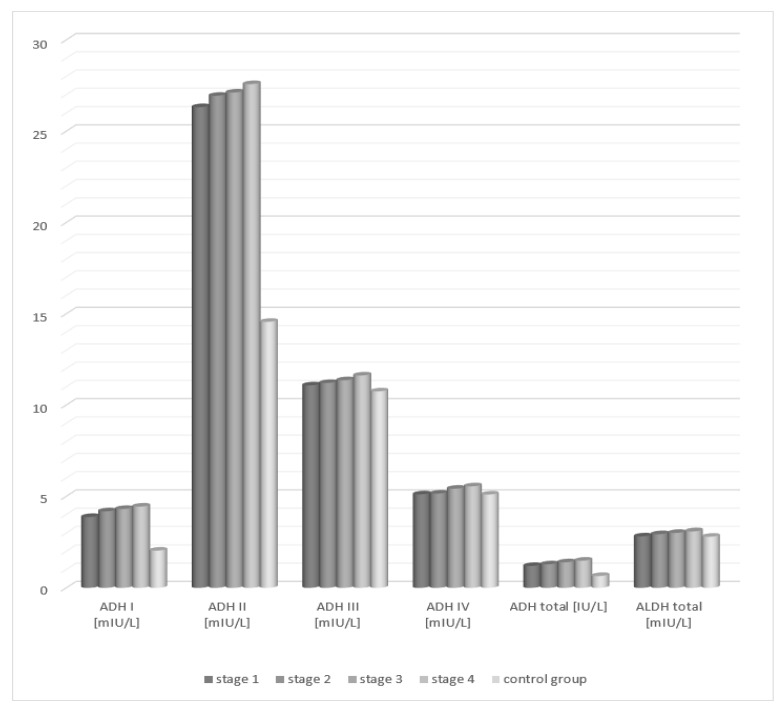
Activity of ADH and ALDH in the serum of patients with alcoholic fatty liver depending on the degree of cirrhosis in accordance with the METAVIR classification.

**Figure 3 medicina-58-00025-f003:**
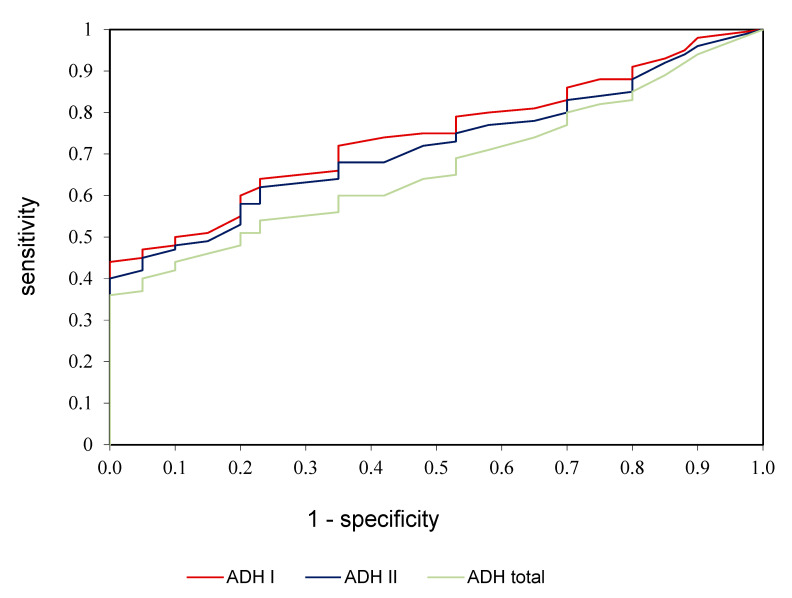
ROC analysis of ADH I, ADH II, and ADH total in patients with alcoholic fatty liver compared to control group.

**Table 1 medicina-58-00025-t001:** Activity of ADH and ALDH in the serum of patients with alcoholic fatty liver.

STUDY GROUPS	ADH I	ADH II	ADH III	ADH IV	ADH Total	ALDH Total
	Median	Median	Median	Median	Median	Median
	Range	Range	Range	Range	Range	Range
	Mean	Mean	Mean	Mean	Mean	Mean
Patients with alcoholic fatty liver (*n* = 38)	4.45	29.21	12.43	5.18	1.41	3.08
	1.25–6.84	11.42–48.76	6.85–20.11	2.54–11.66	0.33–2.82	1.36–6.58
	4.17	26.54	11.03	5.2	1.14	2.84
Control group (*n* = 40)	2.14	15.56	11.75	5.02	0.63	2.8
	0.86–5.36	5.84–22.30	6.86–19.14	2.20–10.92	0.23–1.99	1.22–6.21
	1.85	13.44	10.42	4.98	0.52	2.68
	*p* < 0.01	*p* < 0.01	*p* = 0.362	*p* = 0.514	*p* < 0.001	*p* = 0.551

**Table 2 medicina-58-00025-t002:** Activity of ADH and ALDH in the serum of patients with fatty liver depending on the degree of cirrhosis in accordance with the METAVIR classification.

STUDY GROUPS	ADH I	ADH II	ADH III	ADH IV	ADH Total	ALDH Total
	Median	Median	Median	Median	Median	Median
	Mean ± SD	Mean ± SD	Mean ± SD	Mean ± SD	Mean ± SD	Mean ± SD
1st degree	3.85	26.32	11.08	5.01	1.2	2.72
(*n* = 12)	3.77 ± 2.32	26.07 ± 23.26	11.06 ± 9.45	4.89 ± 4.26	1.08 ± 0.99	2.62 ± 3.21
2nd degree	4.08	26.94	11.21	5.15	1.29	2.84
(*n* = 8)	3.95 ± 2.61	26.48 ± 24.75	11.16 ± 9.22	5.02 ± 4.64	1.11 ± 1.03	2.78 ± 3.64
3rd degree	4.21	27.32	11.36	5.42	1.4	3.01
(*n* = 13)	4.07 ± 2.56	26.85 ± 24.01	11.34 ± 9.02	5.34 ± 4.35	1.26 ± 1.01	2.95 ± 3.12
4th degree	4.44	27.38	11.62	5.45	1.39	3.09
(*n* = 5)	4.33 ± 2.14	27.46 ± 24.53	11.48 ± 9.90	5.30 ± 4.77	1.20 ± 1.05	3.04 ± 3.54
Control group	2.14	14.56	10.75	5.1	0.65	2.7
(*n* = 40)	1.85 ± 1.66	13.44 ± 8.66	10.42 ± 8.34	4.98 ± 4.82	0.52 ± 1.25	2.58 ± 3.41
	*p*^a^ < 0.01	*p*^a^ < 0.01	*p*^a^ = 0.464	*p*^a^ = 0.416	*p*^a^ < 0.01	*p*^a^ = 0.532
	*p*^b^ < 0.01	*p*^b^ < 0.01	*p*^b^ = 0.421	*p*^b^ = 0.347	*p*^b^ < 0.01	*p*^b^ = 0.495
	*p*^c^ < 0.01	*p*^c^ < 0.01	*p*^c^ = 0.376	*p*^c^ = 0.404	*p*^c^ < 0.01	*p*^c^ = 0.566
	*p*^d^ < 0.01	*p*^d^ < 0.01	*p*^d^ = 0.301	*p*^d^ = 0.337	*p*^d^ < 0.01	*p*^d^ = 0.548

*p* ^a^: 1st degree vs. control group, *p* ^b^: 2nd degree vs. control group, *p* ^c^: 3rd degree vs. control group, *p* ^d^: 4th degree vs. control group.

**Table 3 medicina-58-00025-t003:** Receiver operating characteristic (ROC) analysis of ADH I, ADH II, and ADH total of patients with alcoholic fatty liver and the controls.

Parameter	AUC	*p*-Value	Cut-Off Value	Sensitivity (%)	Specificity (%)	95% Confidence Interval
ADH I	0.784	<0.01	5.05	74.56	78.35	1.25–6.84
ADH II	0.758	<0.01	36.81	70.15	75.72	11.42–48.76
ADH total	0.706	<0.001	2.45	66.42	70.33	0.33–2.82

## Data Availability

The full data presented in this study are available on request from the corresponding author.

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
