# Peer review of "Activity of Total Alcohol Dehydrogenase, Alcohol Dehydrogenase Isoenzymes and Aldehyde Dehydrogenase in the Serum of Patients with Alcoholic Fatty Liver Disease"

_medicina, 2021, doi:10.3390/medicina58010025_

Round 1

Reviewer 1 Report

The manuscript (medicina-1500413) entitled " Activity of total Alcohol Dehydrogenase, Alcohol Dehydrogenase Isoenzymes and Aldehyde Dehydrogenase in the Serum of Patients with Alcoholic Fatty Liver Diseaseprovided the interesting results about activity of ADH and ALDH in patients with alcoholic fatty liver disease. Although this paper provided ADH and ALDH data according to METAVIR classification, we need more information of underlying liver function, other baseline characteristics including HBV/HCV infected state, BMI, etc. In addition, we would like to know the histological degree of liver inflammation and its association with ADH or ALDH, and the difference in ADH or ALDH between the control and patient with ALD according to sex and age, respectively. I would like to know how the difference in ADH affects the patient's disease progression or prognosis, and it is thought that this will increase the value of the thesis. If authors will revise the paper according to my comments, it is worth to publish.

Author Response

Thank you for your review. We try to correct our article according to your suggestions and tips.

  1. Infection with HBV or HCV and chronic liver disease was an exclusion criterion. (Materials)
  2. Every patient had a BMI within the normal value (18.5 – 24.9). (Materials)
  3. The diagnosis was performed on the basis of clinical and laboratory investigations ( e.g. transaminases activities).
  4. Gender and age influence ADH levels in general, but the test group and the control group were similar in both cases. Therefore, these factors have been omitted.
  5. ,,It could be supposed, that the high activity of ADH influences the possible development of the disease, as it happens with liver cancer” and this sentence we added in the discussion.

Reviewer 2 Report

The authors measured the activity of total alcohol dehydrogenase (ADH) and its isoenzymes as well as aldehyde dehydrogenase (ALDH) in the serum of patients with alcoholic liver disease. They found that activities of ADH â… , ADH â…¡ and total ADH in patients with alcoholic liver disease were significantly higher than those in healthy control subjects.

Comments

1) The ‘Conclusion’ section mentioned ‘This study identified the possibilities for the application of alcohol dehydrogenase (and its isoenzymes) as well as aldehyde dehydrogenase in the diagnosis of liver cirrhosis’. However, the high activities of the ADH â… , ADH â…¡ and total ADH were observed independent of the degree of the liver fibrosis stages. (Table 2). Kindly make the description consistent with the other parts in the manuscript. The description in the ‘Abstract’ section (The obtained results imply the diagnostic usefulness of the determination of AHD total, ADH I and ADH II activity in the serum of patients with alcoholic fatty liver) should be correct in the paper.

2) The activities of ADH and its isozyme were high in patients with the alcoholic liver disease (Table 2 and Figure 2). The increased activity was commonly observed patients with all fibrotic stages, and seems to be a characteristic of patients with alcoholic fatty liver disease. They should assess whether the activities of ADH (and its isozyme) alter in line with the degree of hepatic steatosis (a histological manifestation of alcoholic liver disease). In addition, they should examine whether the activities of ADH (and its isozyme) were related to the amount of alcohol intake.

3) In relation to my comment No.2, I would like to know the data of patients who stopped drinking habit. Does the activity of the AHD decrease to the level of that in healthy subjects? If previous drinkers still show the high activity, the diagnostic performance may be affected.

Author Response

Thank you for your review. We try to correct our article according to your suggestions and tips.

Point 1.

 A description of every part of the manuscript has been unified (In the first sentence of conclusion ,,cirrhosis" to be changed to ,,steatosis").

Point 2.

We observed the increased activity of total ADH and its isoenzymes (ADH I, ADH II), which correlate with the level of fatty liver, but the difference wasn’t statistically significant.

Every patient has been overusing alcohol for more than 10 years. There weren't any cases of successful alcohol withdrawal recorded. It was a homogenous group when it comes to alcohol consumption.

Point 3.

There weren't any cases of successful alcohol withdrawal recorded.

Round 2

Reviewer 2 Report

The authors responded to the comments. I only have some minor comments.

Minor comments:

1) Line 293-294: They described ‘According to Enomoto et al.,….’, but no reference paper was shown in this sentence. If they should cite the paper by Enomoto N. et al, (Reference No.24), kindly cite here and change the reference numbers accordingly.

2) They should note that the diagnostic performances shown in the Table 3 were moderate, but not very high (AUROCs: 0.706-0.784).

Author Response

Thank you for your revision. We corrected our article according to your suggestions.

All corrections has been marked by red font.

This manuscript is a resubmission of an earlier submission. The following is a list of the peer review reports and author responses from that submission.